# Is Conservative Treatment Better than Surgical Treatment for Basal Ganglia Hemorrhage in a Conventionally Non-Surgical Indication Group with Poor Motor Function?

**DOI:** 10.3390/jcm11102942

**Published:** 2022-05-23

**Authors:** Chan-Hee Park, Min-Gyu Lim, Hyunwoo Jung, Jae-Gyeong Jeong, Seong-Hyun Park, Ae-Ryoung Kim, Tae-Du Jung

**Affiliations:** 1Department of Rehabilitation Medicine, Kyungpook National University Hospital, Daegu 41944, Korea; chany9090@gmail.com (C.-H.P.); ekalsrbsla@naver.com (M.-G.L.); hwjung_87@naver.com (H.J.); cloud90524@naver.com (J.-G.J.); ryoung20@hanmail.net (A.-R.K.); 2Department of Neurosurgery, School of Medicine, Kyungpook National University, Daegu 41944, Korea; nsdoctor@naver.com; 3Department of Rehabilitation Medicine, School of Medicine, Kyungpook National University, Daegu 41944, Korea

**Keywords:** basal ganglia hemorrhage, stroke rehabilitation, intracerebral hemorrhage score

## Abstract

*Background:* The treatment options for basal ganglia intracerebral hemorrhage (ICH), either surgical or conservative, remain controversial. A previous study developed “A new modified ICH (MICH) score” that suggests which treatment option will be better for basal ganglia ICH. According to this scoring system, a MICH score of 0 or 1 indicates that conservative treatment is better than surgical treatment. The purpose of this study was to determine whether conservative treatment is still a better option for a basal ganglia hemorrhage in patients with MICH scores of 0 and 1, with initial poor motor grades. *Methods:* This retrospective study was comprised of 41 patients with a spontaneous basal ganglia hemorrhage. These patient groups had no previous brain lesions, their initial MICH score was 0 or 1, and the manual muscle test (MMT) of their hand was grade 2+ or lower in the initial evaluation. All patients were transferred to the Department of Rehabilitation Medicine and received rehabilitation treatment. Ten patients had an operative intervention, which was burr-hole aspiration or craniotomy with hematoma removal. The control group included 31 patients who underwent conservative treatment. Outcome evaluations used the Berg Balance Scale (BBS), Modified Barthel Index (MBI), and Brunnstrom recovery stage (BRS) which were evaluated at initial and regular follow-ups after 1, 3, and 6 months. We defined an improvement state if the BRS of their hand became 4 or more in 6 months, which means an escape from the synergic pattern. *Results:* Demographic characteristics showed no significant differences in age, sex, hemiplegic side, initial GCS score, presence of IVH and hydrocephalus, or distribution of MICH scores 0 and 1. There was only a significant difference in the distribution of hematoma volume between groups of less than 20 mL and groups from 21 to 50 mL (*p* = 0.049). There were no significant differences in MBI values in 1 month or improvement of BRS of their hand in 6 months between the two groups. *Conclusions:* Even in the group classified as predominantly conservative in basal ganglia hemorrhage patients, if the initial muscle strength is low, it is necessary to reconsider whether conservative treatment is superior to surgical treatment in terms of functional improvement.

## 1. Introduction

The treatment options for a basal ganglia intracerebral hemorrhage (ICH), either surgical or conservative, remain controversial. Previous studies have indicated which one is a better choice for treatment [1,2,3,4,5]. One of those studies developed “A new modified intracerebral hemorrhage (MICH) score” that suggests which treatment option will be better for basal ganglia ICH in the emergency department by grading with a MICH score [6,7]. This scoring system consists of the initial Glasgow Coma Scale (GCS) and hematoma volume and is accompanied by intraventricular hemorrhage (IVH) or hydrocephalus (Table 1). This scoring system used outcome evaluations, 6-month mortality, the Glasgow Outcome Scale (GOS), and the Barthel Index. The GOS and Barthel Index were scored at a 1-year follow-up. It defined good functional outcomes as GOS ≥ 4 or a Barthel Index ≥ 55 [8]. In this scoring system, a MICH score of 0 or 1 indicates that conservative treatment is better than surgical treatment to preserve neurologic function. Surgery is recommended for patients with midlevel MICH scores to obtain better functional outcomes (MICH = 2) and to reduce mortality (MICH = 3 or 4). There are no indications for surgery in patients with a MICH score of 5. However, according to our clinical experience, this method has a limitation in not reflecting the initial muscle strength ability. Even in patients with basal ganglia cerebral hemorrhage with reduced initial muscle strength, the patient group treated with conservative treatment, known to have a generally good prognosis, did not tend to show superior results compared to surgical treatment. Therefore, the purpose of this study was to determine whether conservative treatment is still a better option for a basal ganglia hemorrhage in patients with MICH scores of 0 and 1 with initial poor motor grades.

## 2. Materials and Methods

### 2.1. Participants

This retrospective study comprised 41 patients with a spontaneous basal ganglia hemorrhage who were admitted to the Department of Rehabilitation Medicine of Kyungpook National University Hospital from 2013 to 2020. The GCS was calculated from neurologic examinations in the emergency department. The hematoma volume from the initial computed tomography scan was calculated as ABC/2. In this formula, A is the greatest diameter on the largest hemorrhage slice, B is the diameter perpendicular to A, and C is the approximate number of axial slices with hemorrhage multiplied by the slice thickness.

The inclusion criteria were as follows: (1) patients were diagnosed with the first episode of spontaneous basal ganglia hemorrhage with brain computed tomography or magnetic resonance imaging; (2) patients were transferred to the Department of Rehabilitation Medicine after acute management of brain hemorrhage; (3) patients had a Brunnstrom recovery stage (BRS) of 3 or less with initial hand flexor muscle strength of 2+ or less; and (4) patients were rated with a MICH score of 0 or 1 for which conservative treatment is known to prevail. The exclusion criteria were as follows: (1) patients previously had a history of head trauma, brain metastatic disease, or central nervous system infections and (2) patients had severe musculoskeletal problems such as a fracture.

Ten patients had an operative intervention, which was a burr-hole aspiration or craniotomy with hematoma removal [9,10,11,12]. On the other hand, the control group included 31 patients who underwent conservative treatment including blood pressure control and preventive antiepileptic medication. All 41 patients were transferred for rehabilitation. All of them received rehabilitation programs consisting of physical therapy (PT) and occupational therapy (OT). The PT program had sitting, standing, and gait balance training, gait endurance training, lower extremities strengthening, and range-of-motion exercises. In patients with sensory impairment, it also included visual feedback or eye and body coordination. The OT program had gross hand or fine motor training, in-hand manipulation training, activity training for daily living, upper extremities strengthening, and range-of-motion exercises. Cognitive rehabilitation and dysphagia treatment were included if patients had cognitive impairment or dysphagia, respectively. The PT and OT programs were performed two times a day for 30 min per session. Both rehabilitation therapies were based on the Guide of Physical Therapist Practice (1). This study was approved by the Institutional Review Board of Kyungpook National University Hospital (No. 2022-01-007-002).

### 2.2. Outcome Measures

Outcome evaluations used the Berg Balance Scale (BBS), Modified Barthel Index (MBI), Motricity Index (MI), and Brunnstrom recovery stage (BRS). These outcomes were evaluated at initial and regular follow-ups after 1, 3, and 6 months. We estimated motor function and ambulatory ability using the BBS, BRS, and MI [13,14,15,16]. The BBS is useful for looking at changes, including improvements in functional motor abilities, after stroke. The BRS is an evaluation tool consisting of six steps used in stroke patients and evaluates the upper extremities, hands, and lower extremities [17]. Similarly, this evaluation index evaluates muscle function and consists of three parts: the arms, legs, and trunk. In particular, we measured the change in the evaluation index score after 6 months; at this time of evaluation, a score of 4 or higher indicating escape of the synergistic pattern was set as an improved state. We set this up because we needed patients to be able to release our hands in order to use them functionally. We also considered the hand as an indicator of damage to the corticospinal tract, as this tract is responsible for most of the hand control. Considering these points, we set this score as a cut-off point for improving hand function. In addition, MBI was measured for assessing performance in activities of daily living (ADL) [18]. The MBI is classified into 0 to 100 scores according to dependence. All the measurements were assessed by expert physiotherapists who specialized in assessments and physical evaluation. We compared the outcomes in each evaluation between the two groups using a statistical method. Data were analyzed by chi-square test, Fisher’s exact test, and Mann–Whitney test, using SPSS version 15.0 (SPSS Inc., Chicago, IL, USA).

## 3. Results

According to the demographic characteristics of the two groups, there were no significant differences in age, sex, hemiplegic side, initial GCS score, presence of IVH and hydrocephalus, and distribution of MICH scores 0 and 1. However, there was a significant difference in the distribution of hematoma volume between groups of less than 20 mL and groups from 21 to 50 mL (*p* = 0.049) (Table 2). When comparing the two groups, there was no significant difference between the BRS improvement 6 months after onset (*p* = 0.929) or the MBI score 1 month after onset (*p* = 0.200) (Table 3).

## 4. Discussion

According to our study, even in the group in which conservative treatment is known to be dominant when the initial muscle weakness is severe, we need to consider whether the existing tendency that conservative treatment prevails is correct. We think that this study is meaningful in that it shows a different direction from previous studies that indicate conservative treatment prevails, although there are some limitations. In this regard, it is difficult to say that conservative treatment is superior to surgical treatment in the case of severe initial muscle weakness, even in the patient group where conventional conservative treatment is prioritized.

From this point of view, we would like to note that large-scale and long-term additional studies are needed, considering not only the existing factors such as GCS, the size of the hematoma, and the presence of IVH or hydrocephalus but also the early significant muscle weakness [19,20,21,22,23,24,25,26].

This study is meaningful in that it can suggest various viewpoints by specifying the inclusion criteria in the treatment of basal ganglia hemorrhage patients where there has been difficulty in reaching a consensus between conservative and surgical treatment.

However, our study also has limitations. The number of this study group is limited because it is a retrospective study of cases in which surgical treatment was performed in a situation in which conservative treatment is known to be superior to surgical treatment. Although there were not many patient groups who met all of these conditions, this study was conducted to question the existing treatment methods for these patients. Additionally, since there is an ethical problem in prospectively recruiting a surgical group for research purposes in a situation where conservative treatment is preferred, we proceeded with the idea that this research method is the best in the current situation. For this reason, the non-inferiority test could not be performed because the sample was too small to have normality. In addition, a normality test was performed to compare MBI scores, but a non-parametric Mann–Whitney test was performed because a normal distribution was not followed, and the mean and standard deviation between the two groups could not be compared. In a future follow-up study, if normality through a sufficient sample size can be obtained, more meaningful comparisons can be made. Additionally, regardless of whether conservative treatment or surgery was performed, all of the groups that improved were relatively poorly followed, so there was a limitation in the longitudinal aspect. For this reason, there was a limit to the data that could be compared because the completion of the longitudinal outcome data was limited.

In addition, as a result of the statistical comparison of hematoma volume between groups in Table 2, a statistically significant *p*-value of 0.049 (<0.05) was shown. This indicates that the proportion of patients with a lower hematoma volume in the conservative treatment group was significantly higher despite the relatively identical MICH score of 0 or 1, suggesting that the two groups had an imbalance. In addition, these results should be considered noting that the hematoma volume in a relatively mild patient group, such as a MICH score of 0 or 1, had an effect on determining the treatment method. Therefore, in order to clearly analyze these effects in future studies, it is necessary to control for the imbalance between groups in hematoma volume.

## 5. Conclusions

According to our study, it is necessary to reconsider whether conservative treatment is superior to surgical treatment in terms of functional improvement if initial muscle strength is low, even in the group classified as predominant in basal ganglia hemorrhage patients. In addition, it is necessary to carefully examine whether there is a decrease in initial muscle strength when determining the treatment direction.

## Figures and Tables

**Table 1 jcm-11-02942-t001:** A new modified intracerebral hemorrhage (MICH) score.

Component	ICH Score Points
GCS score	
15–13	0
12–5	1
4–3	2
ICH volume, mL	
≤20	0
21–50	1
≥51	2
IVH or hydrocephalus	
No	0
Yes	1
Total MICH score	0–5

Glasgow Coma Scale (GCS) score indicates GCS score presentation (or after resuscitation); intracerebral hemorrhage (ICH) volume, volume on initial computed tomography calculated using ABC/2 method; and intraventricular hemorrhage (IVH), presence of any IVH on initial computed tomography.

**Table 2 jcm-11-02942-t002:** Characteristics of patients.

	Conservative (*n* = 31)	Operative (*n* = 10)	*p*-Value
Age (mean)	54.52 (32–85)	54.80 (36–76)	0.709
M/F	22/9	6/4	0.698
Hemiplegic side (R/L)	13/18	6/4	0.064
GCS (mean)	13.39	13.39	0.940
IVH (*n*, %)	3 (9.6%)	0 (0%)	0.564
Hydrocephalus (*n*, %)	0 (0%)	0 (0%)	
Hematoma volume (mL)			
≤20	24 (77.41%)	4 (40%)	0.049 *
21–50	7 (22.58%)	6 (60%)	
≥51	0 (0%)	0 (0%)	
MICH score (*n*, %)			
0	14 (45.16%)	3 (30%)	0.480
1	17 (54.83%)	7 (70%)	

*p*-value, comparison between the conservative and operative groups by chi-square test and Fisher’s exact test (*, *p*-value < 0.05); M, male; F, female; R, right; L, left; GCS, Glasgow Coma Scale; IVH, intraventricular hemorrhage; MICH, modified intracerebral hemorrhage.

**Table 3 jcm-11-02942-t003:** Distribution according to whether there was improvement between the operated and conservatively treated group.

	Conservative (*n* = 31)	Operative (*n* = 10)	*p*-Value
Brunnstrom recovery stage			
Improvement	16	5	0.929 *
No improvement	15	5	
MBI score after 1 month (mean ± SD)	34.68 ± 22.401	26.80 ± 27.462	0.200 ^†^

*p*-value, comparison between the conservative and operative groups by chi-square test * and Mann–Whitney test ^†^; Improvement: Brunnstrom recovery stage (BRS) in hand ≥4 at 6 months follow-up; No improvement: BRS in hand <4 at 6 months follow-up; MBI: Modified Barthel Index; SD: Standard Deviation.

## Data Availability

Available upon reasonable request.

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
