# Peer review of "Is Conservative Treatment Better than Surgical Treatment for Basal Ganglia Hemorrhage in a Conventionally Non-Surgical Indication Group with Poor Motor Function?"

_jcm, 2022, doi:10.3390/jcm11102942_

Round 1
Reviewer 1 Report
This study was an analysis of outcomes in patients with low (good grade) MICH score for motor recovery in patients with basal ganglia haemorrhage. The authors compared the outcomes between surgery treated patients (10 patients) and medical management (31 patients). There was no difference in the motor recovery between the two groups on follow up, up to 6 months after ictus.
The study has major limitations.
- The cohort is small
- Surgical evacuation for deep haemorrhage is still under investigation and its role in ICH management remains controversial. There is information lacking regarding timing of the surgery in relation to the ictus of ICH. The number of surgical inverventions is small also. However all patients had low moderate ICH volume, perhaps not surprising there is no difference between the two treatment arms.
- Not specifically mentioned, did all the patients survive their ICH?
Author Response
This study was an analysis of outcomes in patients with low (good grade) MICH score for motor recovery in patients with basal ganglia hemorrhage. The authors compared the outcomes between surgery treated patients (10 patients) and medical management (31 patients). There was no difference in the motor recovery between the two groups on follow up, up to 6 months after ictus.
The study has major limitations.
Point 1: The cohort is small
Response 1: We appreciate your comment. As pointed out, we believe that the small number of subjects is one of the major limitations of this study. As mentioned in the Discussion, the number of this study group is limited because it is a retrospective study of cases in which surgical treatment was performed in a situation in which conservative treatment is known to be superior to surgical treatment. In other words, although the MICH score is 0 or 1, which is known to be the dominant treatment for conservative treatment, patients with severe initial muscle weakness were selected and among them, patients who received surgical treatment were targeted. Although there were not many patient groups who met all of these conditions, this study was conducted to question the existing treatment methods in these patients. Additionally, since there is an ethical problem in prospectively recruiting a surgical group for research purposes in a situation where conservative treatment is preferred, we proceeded with the idea that this research method is the best in the current situation.
Therefore, the following sentences have been added to the text for detailed explanation.
(Page 5) Discussion
The number of this study group is limited because it is a retrospective study of cases in which surgical treatment was performed in a situation in which conservative treatment is known to be superior to surgical treatment. Although there were not many patient groups who met all of these conditions, this study was conducted to question the existing treatment methods in these patients. Additionally, since there is an ethical problem in prospectively recruiting a surgical group for research purposes in a situation where conservative treatment is preferred, we proceeded with the idea that this research method is the best in the current situation.
Point 2: Surgical evacuation for deep hemorrhage is still under investigation and its role in ICH management remains controversial. There is information lacking regarding timing of the surgery in relation to the ictus of ICH. The number of surgical interventions is small also. However, all patients had low moderate ICH volume, perhaps not surprising there is no difference between the two treatment arms.
Response 2: We appreciate your comment. The onset of ICH that you pointed out and the timing of surgery are detailed in the table below. However, in this study, as a comparative study between surgical treatment and conservative treatment, comparison within the surgical treatment group was not performed. As for the comparison within the surgical treatment group you mentioned, it seems that it can be implemented after recruiting more surgical treatment cases in the future.
|
Table. Time from onset to operation of operation group |
|
|
Patient No. |
Time from onset to operation (hours) |
|
1 |
41 |
|
2 |
15 |
|
3 |
18 |
|
4 |
13 |
|
5 |
10 |
|
6 |
23 |
|
7 |
24 |
|
8 |
32 |
|
9 |
21 |
|
10 |
29 |
|
Average |
24.4 |
On the other hand, the role of surgery in the treatment of spontaneous basal ganglia hemorrhage remains a matter of debate. So, as pointed out, this study is to conduct a study that can help in determining the treatment method by comparing the group that underwent surgical treatment and the group that underwent conservative treatment in the group where conservative treatment is known to be relatively superior. Additionally, stereotactic aspiration, craniotomy and endoscopic surgery have been suggested as recent surgical techniques for basal ganglia hemorrhage. In our study, 8 out of 10 patients underwent burr-hole aspiration and 2 patients underwent craniotomy.
Both groups in the surgical or conservative treatment group show low to moderate ICH volume as you said. So, you said that it is not surprising that there is no difference in outcomes between the two groups, but I think the opposite is true. This study is a retrospective study of cases in which surgical treatment was performed in a group with a MICH score of 0 or 1, which is known to be superior to conservative treatment in terms of outcome. The authors thought that the absence or small difference between the two types of treatment provides a rationale for raising the question of whether it can be said that conservative treatment is superior.
Point 3: Not specifically mentioned, did all the patients survive their ICH?
Response 3: We appreciate your comment. All 41 patients survived and were transferred for rehabilitation.
(Page 3) Materials and Methods
All 41 patients survived and were transferred for rehabilitation. All patients of them received rehabilitation program consisting of physical therapy (PT) and occupational therapy (OT).
Reviewer 2 Report
This study compared the clinical outcome between two treatments group (operative intervention vs. conservative treatment) in basal ganglia hemorrhage patients with MICH scores 0 and 1.
Several comments to the authors:
- Does the MBI score data come from a normal distribution? Is it better to compare the median with a non-parametric test? Could the author please give more detail about the statistical analysis, e.g., the statistical Signiant level?
- In the abstract and methods section mentioned: The outcome evaluations used Berg Balance Scale (BBS), Modified Barthel Index 23 (MBI), and Brunnstrom recovery stage (BRS), which were evaluated at initial and regular follow-up after 1, 3, 6 months. However, the result section and Table 3 only provide BRS at 6 months and MBI at 1 month. What are the other results on other follow up time points?
- Without a table comparing operative intervention vs. conservative treatment followed by formal statistical analyses possibly requiring propensity weighting in case of severe imbalance, it is difficult to be sure the findings are valid. The conservative treatment group had a significant higher of portion patients with smaller hematoma volume (less than 20 mL). At least the author should discuss the limitation of the imbalance between two groups.
- The operative treatment group only includes ten patients; it is difficult to ensure the finding is valid with such a small sample size.
Author Response
This study compared the clinical outcome between two treatments group (operative intervention vs. conservative treatment) in basal ganglia hemorrhage patients with MICH scores 0 and 1.
Several comments to the authors:
Point 1: Does the MBI score data come from a normal distribution? Is it better to compare the median with a non-parametric test? Could the author please give more detail about the statistical analysis, e.g., the statistical Signiant level?
Response 1: We appreciate your comment. The normal distribution was not followed as a result of performing a normality test on the group divided by the MBI score and the presence or absence of surgery. So, as you said, we performed the Mann-Whitney test, which is a non-parametric test. However, methods and specific descriptions of statistical techniques used in Table 3 were not described, so they have been modified as follows.
On the other hand, the Mann-Whitney test is a rank sum test, which mixes data from two groups, sorts them in order of size, assigns ranks, and calculates the sum of ranks for each group to test whether the size of the rank sum of the two groups is statistically different. In this process, intrinsic values of the original data remain only in rank, and all are lost and do not affect the analysis, so the mean and standard deviation of the two groups are excluded as they are considered to have no meaning in the hypothesis test.
In this respect, it is expected that more meaningful results can be obtained if a sufficient sample size is secured in future studies and statistical analysis can be performed in the state of normality.
(Page 5) Discussion
In addition, normality test was performed to compare MBI scores, but a non-parametric Mann-Whitney test was performed because a normal distribution was not followed, and the mean and standard deviation between the two groups could not be compared. In a future follow-up study, if normality through sufficient sample size can be obtained, more meaningful comparisons can be made.
Point 2: In the abstract and methods section mentioned: The outcome evaluations used Berg Balance Scale (BBS), Modified Barthel Index 23 (MBI), and Brunnstrom recovery stage (BRS), which were evaluated at initial and regular follow-up after 1, 3, 6 months. However, the result section and Table 3 only provide BRS at 6 months and MBI at 1 month. What are the other results on other follow up time points?
Response 2: We appreciate your comment. As mentioned in the limitation of the discussion, follow-up was not completed regularly due to being hospitalized at another hospital or due to various personal circumstances. So, the MBI score of the first month and the BRS after 6 months, when all the subjects were completely evaluated, were presented in the manuscript. Although not all of them were evaluated, the data investigated during each follow-up period and the results of statistical analysis are added below for supplementary explanation. In the manuscript, only the statistical analysis results based on the completed data are written first.
|
Table 4. Distribution according to whether or not there was improvement between the operated and conservatively treated group |
|||
|
|
Conservative |
Operative |
p-value |
|
Brunnstrom recovery stage after 1 month |
N=23 |
N=10 |
|
|
Improvement |
7 |
2 |
0.536 |
|
No improvement |
16 |
8 |
|
|
Brunnstrom recovery stage after 3 months |
N=25 |
N=8 |
|
|
Improvement |
7 |
4 |
0.251 |
|
No improvement |
18 |
4 |
|
|
Brunnstrom recovery stage after 6 months |
N=31 |
N=10 |
|
|
Improvement |
16 |
5 |
0.929 |
|
No improvement |
15 |
5 |
|
|
p-value, comparison between the conservative and operative groups by chi-square test and Fisher’s exact test; Improvement; Brunnstrom recovery stage (BRS) in hand ≥4 at each follow-up, No improvement; BRS in hand <4 at each follow-up |
|||
Table 5. Comparison of MBI scores between the operated and conservatively treated group
|
|
Conservative |
Operative |
p-value |
|
MBI score after 1 month |
N=31 |
N=10 |
0.200 |
|
34.68 ± 22.401 |
26.80 ± 27.462 |
||
|
MBI score after 3 month |
N=26 |
N=8 |
0.253 |
|
55.96 ± 19.49 |
66.36± 35.20 |
||
|
MBI score after 6 month |
N=16 |
N=6 |
0.112 |
|
65.2± 17.87 |
81.83± 24.63 |
p-value, comparison between the conservative and operative groups by Mann-Whitney test; MBI; Modified Barthel Index, SD; Standard Deviation.
Point 3: Without a table comparing operative intervention vs. conservative treatment followed by formal statistical analyses possibly requiring propensity weighting in case of severe imbalance, it is difficult to be sure the findings are valid. The conservative treatment group had a significant higher of portion patients with smaller hematoma volume (less than 20 mL). At least the author should discuss the limitation of the imbalance between two groups.
Response 3: We appreciate your comment. As shown in Table 2, we compared demographics between the two groups. As mentioned, there was a significant difference between the groups with a p-value of 0.049 as a result of performing the chi-square test only on the ICH volume between the two groups. This content is described in Results, and the following sentences have been added to the discussion section for further explanation.
(Page 3) Results
However, there was a significant difference in the distribution of hematoma volume between groups less than 20 and groups from 21 to 50 (p=0.049) (Table 2).
(Page 5) Discussion
In addition, as a result of statistical comparison of hematoma volume between groups in Table 2, a statistically significant p-value of 0.049 (< 0.05) was shown. This indicates that the proportion of patients with a lower hematoma volume in the conservative treatment group was significantly higher even though the relatively identical MICH score was 0 or 1, suggesting that the two groups had imbalance. In addition, these results can be considered that hematoma volume in a relatively mild patient group such as a MICH score of 0 or 1 had an effect on determining the treatment method. Therefore, in order to clearly analyze these effects in future studies, it is thought that it is necessary to control for imbalance between groups in hematoma volume.
Point 4: The operative treatment group only includes ten patients; it is difficult to ensure the finding is valid with such a small sample size.
Response 4: We appreciate your comment. As pointed out, we believe that the small number of subjects is one of the major limitations of this study. As mentioned in the Discussion, the number of this study group is limited because it is a retrospective study of cases in which surgical treatment was performed in a situation in which conservative treatment is known to be superior to surgical treatment. In other words, although the MICH score is 0 or 1, which is known to be the dominant treatment for conservative treatment, patients with severe initial muscle weakness were selected and among them, patients who received surgical treatment were targeted. Although there were not many patient groups who met all of these conditions, this study was conducted to question the existing treatment methods in these patients. Additionally, since there is an ethical problem in prospectively recruiting a surgical group for research purposes in a situation where conservative treatment is preferred, we proceeded with the idea that this research method is the best in the current situation.
Therefore, the following sentences have been added to the text for detailed explanation.
(Page 5) Discussion
The number of this study group is limited because it is a retrospective study of cases in which surgical treatment was performed in a situation in which conservative treatment is known to be superior to surgical treatment. Although there were not many patient groups who met all of these conditions, this study was conducted to question the existing treatment methods in these patients. Additionally, since there is an ethical problem in prospectively recruiting a surgical group for research purposes in a situation where conservative treatment is preferred, we proceeded with the idea that this research method is the best in the current situation.
Round 2
Reviewer 1 Report
Nil further